# The Contribution of Precision Nutrition Intervention in Subfertile Couples

**DOI:** 10.3390/nu17010103

**Published:** 2024-12-30

**Authors:** Jéssica Monteiro, Manuel Bicho, Ana Valente

**Affiliations:** 1Applied Nutrition Research Group (GENA), Nutrition Lab, Egas Moniz Center for Interdisciplinary Research (CiiEM), Egas Moniz School of Health & Science, 2829-511 Caparica, Portugal; jessica.valente.monteiro15@gmail.com; 2Ecogenetics and Human Health Research Group, Associate Laboratory TERRA, ISAMB—Institute of Environmental Health, Lisbon School of Medicine, University of Lisbon, 1649-028 Lisbon, Portugal; manuelbicho@medicina.ulisboa.pt; 3Institute of Scientific Research Bento da Rocha Cabral, Calçada Bento da Rocha Cabral 14, 1250-012 Lisbon, Portugal

**Keywords:** precision nutrition, nutrigenetics, subfertility, couples, Mediterranean pattern, obesity, macronutrients, micronutrients

## Abstract

Background/Aim: Subfertility is characterized by a decrease in reproductive efficiency, which can result in delayed pregnancy, and affects one in six individuals during their lifetime. The present narrative review aims to evaluate the contribution of precision nutrition to changes in fertility in subfertile couples. Methods: The literature review was carried out through bibliographic research in the PubMed, Scopus, SciELO and Google Scholar databases. The following search criteria were applied: (1) original articles and narrative, systematic or meta-analytic reviews, and (2) the individual or combined use of the following keywords: “genetic variation”, “nutrigenetics”, “precision nutrition”, “couple’s subfertility”, and “couple’s infertility”. A preliminary reading of all the articles was carried out, and only those that best fit the themes and subthemes of the narrative review were selected. Results: Scientific evidence suggests that adherence to a healthy diet that follows the Mediterranean pattern is associated with increased fertility in women and improved semen quality in men, better metabolic health and reduced levels of inflammation and oxidative stress, as well as maintaining a healthy body weight. The integration of different tools, such as nutrigenetics, predictive biochemical analyses, intestinal microbiota tests and clinical nutrition software, used in precision nutrition interventions can contribute to providing information on how diet and genetics interact and how they can influence fertility. Conclusions: The adoption of a multidisciplinary and precision approach allows the design of dietary and lifestyle recommendations adapted to the specific characteristics and needs of couples with subfertility, thus optimizing reproductive health outcomes and achieving successful conception.

## 1. Introduction

Infertility is characterized by the World Health Organization (WHO) and the International Committee for Monitoring Assisted Reproductive Technologies (ICMART) as a disease of the reproductive system, defined by the inability to become pregnant after at least 12 months of regular, unprotected sexual intercourse [1]. Subfertility, however, is characterized by a decrease in reproductive efficiency, which can result in delayed pregnancy [2]. According to a report published by the World Health Organization, based on data from 1990 to 2021, infertility is a public health problem that affects approximately 17.5% of the adult population, with one in six individuals in the world suffering from subfertility throughout their lives [3], affecting around 50 million couples worldwide [4,5]. Regarding the causes of infertility, 20–35% of diagnoses are attributed to physiological causes in women, approximately 20–30% in men and 25–40% result from problems in both sexes. However, around 10–20% of infertility cases have unknown causes, making it impossible to implement treatment strategies [6]. Female-related causes of infertility are often associated with ovulatory dysfunction, obstruction of the fallopian tubes, uterine or peritoneal diseases, problems with the reproductive system or chronic diseases [7]. In men, the main causes are related to abnormalities in sperm count, morphology and motility [8], with oxidative stress identified as a preponderant factor for sperm quality and fertilization potential [9]. Assisted reproductive technology (ART) treatments and procedures have emerged as one of the main approaches for couples facing difficulties conceiving. However, the high emotional, physical, financial and geographic impact associated with therapy can be significant, especially due to the high incidence of unsatisfactory results [10,11]. In recent years, there has been a growing interest in identifying modifiable risk factors that may affect couples’ fertility and in personalized nutrition interventions for both members of the couple, with the aim of altering subfertility (Figure 1) [12]. Although there is growing acceptance that nutrition may be related to the reproductive performance of both sexes and a strong link exists between unhealthy eating habits and infertility [13], international guidelines for couples of reproductive age are not yet available [11]. Recently, new areas of scientific knowledge, such as nutrigenetics and intestinal microbiota, have been used in personalized nutrition interventions, particularly in the context of promoting health and well-being, with a potential influence on the modification of subfertility in couples (Figure 1) [14]. Thus, precision nutrition has been described as a promising strategy to improve couples’ fertility through the application of a multidisciplinary approach, with the integration of different elements, some more traditional, such as anthropometry and food consumption, and others from new areas of knowledge, such as nutrigenetics, intestinal microbiota and predictive biochemical parameters [12,13]. The present narrative review aims to evaluate the contribution of precision nutrition in altering fertility in subfertile couples.

## 2. Methodology

The literature review was carried out through bibliographic research in the PubMed, Scopus, SciELO and Google Scholar databases. The following search criteria were applied: (1) original articles and narrative, systematic or meta-analytic reviews, and (2) the individual or combined use of the following five keywords: “genetic variation”, “nutrigenetics”, “precision nutrition”, “couple’s subfertility”, and “couple’s infertility”. A preliminary reading of all the articles was carried out, and only those that best fit the themes and subthemes of the narrative review were selected. Some articles were used to obtain results associated with the description of risk factors for infertility/subfertility, and others were used to write the results of point 2, “Optimizing Couples fertility—Precision Nutrition”, and point 3, “Personalized nutrition interventions to optimize fertility”. The bibliographic references of the articles obtained as a result of the initial research were also read and analyzed. Of these references, those considered relevant were also used in this narrative review of the scientific literature.

## 3. Results and Discussion

### 3.1. Risk Factors for Couple’s Subfertility

#### 3.1.1. Nutritional Status

##### Obesity, Overweight and Underweight in Women

Inadequate nutritional status and deficiencies in essential nutrients have a significant impact on fertility, affecting a variety of physiological and metabolic processes involved in reproduction in both sexes [15]. The correlation between obesity and reproductive functions has been recognized in the scientific literature [16,17]. Although the adverse effects of obesity on reproductive outcomes are known, and there are several mechanisms proposed to explain how obesity can lead to infertility [18], the description of the exact pathophysiology by which obesity affects the reproductive system requires clarity due to its complex and multifaceted character [19]. Obesity and overweight have a significant impact on reproductive health, especially in women, being linked to higher risks of ovulatory infertility and higher incidence of menstrual dysfunction [20,21]. Furthermore, obese or overweight women have an increased risk of subfecundity and infertility, lower conception rates and an increased likelihood of miscarriage and pregnancy complications (body mass and ovulatory index) [19,20,21,22,23]. The work carried out by Rogers and Mitchell (1952) [24] found that obese women were four times more likely to have menstrual disorders. Similarly, Grodstein et al. (1994) [25] showed that anovulatory infertility was greater in overweight and obese individuals with a body mass index (BMI) greater than 26.9 kg/m^2^. Also, Green et al. (1988) [21] studied the relationship between weight and ovulatory infertility in women and found that there was a relative risk of ovulatory infertility of 2.1 (with a 95% CI of 1.0–4.3) in participants with a weight 120% greater than the ideal weight. Edwards et al. (1994) [26] conducted a nested case–control study of 67,649 nurses and observed that there was an increased risk of ovulatory infertility in women with a BMI of 24 or higher (RR = 1.7, 95% CI = 1.5–1.9). The researchers Van Der Steeg et al. (2007) [20], who evaluated whether obesity affected the chance of a spontaneous pregnancy, found that the probability of a spontaneous pregnancy linearly declined with a BMI over 29 kg/m^2^. Weight loss also has a positive impact on reproductive outcomes in obese and pre-obese women [19]. Several studies [19,22,27] suggest that weight loss in anovulatory obese women can lead to ovulation and the return of fertility. Significant weight reduction in women with a high degree of obesity can normalize plasma hormone concentrations and improve the regularity of menstrual cycles [28]. Excessive thinness is also a risk factor for infertility, favoring the occurrence of anovulatory menstrual cycles and hypothalamic dysfunction [29]. Green et al. (1998) [21] observed that women who weighed less than 15% of their ideal weight and who had not previously conceived had a 4.7 times higher risk of infertility attributed to ovulatory dysfunction. The work developed by William Bates et al. (1982) [27] in a sample of women weighing less than at least 15% of their ideal weight and who were followed up in infertility consultations found that 73% of them managed to get pregnant naturally after adhering to a dietary plan to achieve their ideal body weight. This suggests that a healthy, ideal weight can lead to increased fertility. Thus, a healthy weight in women seems to have a preponderant role in promoting female fertility [19,30].

##### Obesity, Overweight and Underweight in Men

Weight management in men is also essential for adequate fertility. Scientific evidence [31,32] has shown that high body mass index (BMI) is associated with changes in testosterone and estrogen levels as well as in concentrations of sex hormone-binding globulin (SHBG). Despite recent research efforts to explore links between increased or decreased BMI and semen quality, results remain inconclusive and controversial [33]. Although some studies [34,35] have indicated an adverse effect of high BMI on semen quality, others have found no significant association [31,36]. Some scientific studies have shown a consistent association between being overweight or obese and various aspects of sperm health, including sperm concentration [37,38,39], sperm motility [39,40], sperm morphology [36] and total sperm count [37,38,41]. On the other hand, contrasting results were reported by other authors [42,43,44,45,46], who found no significant associations between sperm health and weight. Chavarro et al. (2010) [41] showed that men with a BMI ≥ 35 kg/m^2^ exhibited a decrease in ejaculate volume and a lower total sperm count. Years later, Wang et al. (2017) [34] observed similar results in an investigation of 2384 subfertile men in northern China, where increased BMI was associated with decreased total sperm count and lower sperm concentrations. This result agrees with that of another study carried out in 2006, taken from an agricultural health research project in the United States, in which 1329 couples were evaluated. In this study, it was found that the man’s BMI had a significant impact on the couple’s ability to conceive, being considered an independent risk factor for infertility [47]. Additionally, using the same sample, it was shown that male participants undergoing a 14-week weight loss program had an increase in total sperm count and semen volume, which was one of the few studies to associate sperm with weight loss due to increased levels of testosterone, a globulin linked to sex hormones and anti-Mullerian hormone. However, more studies are needed to determine whether these changes are directly associated with reduced body weight or other factors. Obesity has been consistently linked to erectile dysfunction problems, and there is evidence that weight loss can improve erectile functionality [48,49]. Like being overweight, being underweight has also been associated with sperm health. In the study carried out by Ma et al. (2019) [33], it was observed that being underweight and overweight were significantly associated with lower semen quality. In 2007, Qin et al. [50] studied 990 fertile men for 1 year (2001 to 2002) and concluded that low weight was associated with a lower sperm concentration and count. Previously, Jensen et al. (2004) [51] carried out a study on 1558 young Danish men and observed that participants with a BMI < 20 kg/m^2^ had a significant reduction in sperm concentration and total sperm count. The recent meta-analysis carried out by Guo et al. (2019) [52] concluded that there was robust scientific evidence that low male BMI and semen quality are associated, suggesting that low weight is a risk factor for infertility. The continued reduction in energy intake and subsequent weight loss results in a gradual decline in prostate fluid, sperm motility and longevity. When weight loss reaches approximately 25% of normal body weight, sperm production ceases completely [29]. Thus, according to current scientific evidence [53,54], inadequate weight in men represents a risk factor for subfertility in couples, and it is crucial that men with obesity, pre-obesity or low weight seek to normalize their weight and BMI, with the aim of improving sperm quality [31,33].

#### 3.1.2. Nutrigenetics

The interaction between nutrigenetics and subfertility reflects the complexity of factors that influence reproductive health. Environmental factors such as contaminants, stress and dietary compounds during the early stages of fetal and postnatal development have been identified as having a significant impact on health and fertility [55]. Food intake and its interaction with other environmental factors can cause epigenetic changes that can activate or deactivate certain genes [56]. Nutrigenetics is the area of genetics that studies the effect of genetic polymorphisms on the absorption and metabolism of micronutrients as well as food choices that affect health [56]. Nutrigenetic tests have been used in recent years with the aim of optimizing the nutritional status of couples with subfertility problems [57,58]. According to scientific evidence, couples in which at least one partner is overweight are more likely to have subfertility problems [33,34,42,59,60]. The identification of specific genetic polymorphisms is useful for understanding gene–nutrient interactions and is a highly valuable tool in personalized nutrition intervention [61] for one or both members of a couple with subfertility. Genetic polymorphisms can influence the levels of micronutrients in the human body, consequently affecting the nutritional status and general health of an individual [62]. Single nucleotide polymorphisms (SNPs) are closely linked to specific phenotypes associated with micronutrient deficiency. These genetic variations can alter the processes of absorption, transport and utilization of nutrients within the body [62]. Considering that nutritional deficiencies can affect fertility [63], it is essential to understand an individual’s genetic predisposition to these deficiencies through the analysis of associated SNPs. With this knowledge, it is possible to identify individuals with a genotype at greater risk of developing several polygenic diseases and presenting subclinical nutritional deficits [62]. Thus, the identification of SNPs that predispose to nutritional deficiencies allows the identification of genetic risks, enabling personalized nutrition intervention [61] and improving the couple’s reproductive health. Thus, the identification of the couple’s genetic polymorphisms can contribute to improving their nutritional status and positively influencing the couple’s fertility.

#### 3.1.3. Chronic Diseases

Chronic non-communicable diseases such as metabolic syndrome, type 2 diabetes mellitus, polycystic ovary and endometriosis, among others, trigger a chronic inflammatory state in the body. This condition can have adverse effects on reproductive health, affecting both men and women [64].

##### Metabolic Syndrome

Metabolic syndrome (MS) can seriously compromise reproductive function [64]. It is characterized by a combination of excess abdominal and visceral fat, dyslipidemia, hypertension and/or insulin resistance [65]. Some studies [66,67] have demonstrated that there is a relationship between MS and increased oxidative stress and systemic pro-inflammatory states. In the field of male reproductive health, studies have shown that high oxidative stress leads to lipid peroxidation of the sperm membrane, causing a decline in sperm motility with impaired sperm–oocyte interaction [68,69]. Total and bioavailable testosterone, as well as globulin linked to sex hormones, have also been positively associated with several MS risk factors [70]. Dyslipidemia is one of the characteristics of MS that can potentially influence semen quality and fertility [65]. Ramirez Torres et al. (2000) [71] studied 106 male partners of infertile couples and found that 65% had dyslipidemia. Another factor associated with MS, and which has been studied in its relationship with serum testosterone levels, is male arterial hypertension. However, the precise mechanism leading to this observed association between high blood pressure and decreased androgen levels is still poorly understood [65].

##### Diabetes Mellitus

Type 2 diabetes mellitus (T2DM) is closely associated with male infertility [72]. Some scientific studies have demonstrated a consistent relationship between diabetes mellitus (DM) and sexual dysfunction [73], especially erectile [74] and ejaculatory dysfunction [75], together with hypogonadism [74]. In a retrospective analysis carried out by La Vignera et al. (2012) [76], a 51% prevalence of subfertility was identified among individuals diagnosed with DM2 in the study. In the same year, Imani et al. [77] concluded that the prevalence of infertility among men with DM2 was 35.1%, representing a higher frequency than in the population without diabetes. According to current knowledge, type 1 diabetes mellitus (DM1) is also a risk factor for male infertility but due to different pathophysiological mechanisms compared to DM2 [73]. DM1 is associated with reduced ejaculate volume and mitochondrial damage, leading to decreased sperm motility. The pathophysiological mechanisms underlying infertility in DM2 are explained by an inflammatory condition with an increased level of oxidative stress, resulting in decreased sperm vitality and increased sperm DNA fragmentation. In individuals with DM2, spermatogenesis can be interrupted and erectile function is impaired, leading to ejaculation disorders [73]. Some epidemiological studies [78,79] have shown results that highlight that men with DM2 are more vulnerable to erectile dysfunction and that the degree of dysfunction is directly related to diabetic neuropathy [80,81]. Additionally, greater insulin resistance in men with T2DM has been associated with reduced testosterone secretion in the testis, influencing Leydig cell function [82,83]. In relation to the female reproductive system, DM2 also appears to have detrimental effects on fertility [84]. According to a recent scientific review [85], T2DM has been associated with reduced fertility rates because of hyperglycemia and the consequent reduction in fertility and increased rates of spontaneous abortions [86]. In a cohort study [87] carried out in a sample of more than 2 million couples, it was found that maternal hyperglycemia before pregnancy was correlated with decreased fecundability in couples.

##### Polycystic Ovary Syndrome

Polycystic ovary syndrome (PCOS) is a prevalent endocrine disorder found in women of reproductive age, with a prevalence between 5 and 15%, and constitutes the main cause of anovulatory female infertility [88,89]. The commonly accepted characterization of PCOS includes the presence of clinical and/or biochemical evidence of androgen excess along with chronic anovulation after excluding specific underlying pituitary or adrenal disorders [90]. PCOS is often characterized by irregular ovulation, hyperandrogenism, polycystic ovarian morphology, obesity and insulin resistance and is mediated by inflammatory cytokines and adipokines [91]. The development of PCOS in women is influenced by factors such as MS and obesity [92]. Several studies [88,93,94,95] have shown that women with PCOS are more likely to develop MetS. There is an apparent overlap in the pathogenesis of PCOS and MetS, possibly due to shared mechanisms involving insulin resistance. In PCOS, hyperinsulinemia and peripheral insulin resistance are fundamental components of metabolic dysfunction. Women with PCOS also have an increased susceptibility to T2DM [96]. About 40% of obese young women with symptoms characteristic of PCOS, such as hyperandrogenism and anovulation, have impaired glucose tolerance [90]. It is estimated that women with PCOS are three to seven times more likely to develop T2D in their lifetime than weight-matched controls [97,98].

##### Endometriosis

Endometriosis, being an estrogen-dependent chronic inflammatory disease, is characterized by the abnormal growth of endometrial tissue outside the uterine cavity [99]. The scientific literature has shown the existence of elevated levels of inflammatory markers (interleukin-6, tumor necrosis factor α and C-reactive protein) in women with endometriosis [100,101]. The inflammatory process has an impact on lipid metabolism, promoting an increase in low-density lipoprotein cholesterol levels and, at the same time, a reduction in high-density lipoprotein cholesterol levels [102,103]. Chronic inflammation caused by endometriosis may play a crucial role in the onset of MetS and DM2 [104], and there is currently scientific evidence that endometriosis is a risk factor for the development of MetS and DM2 [99,101,105]. Chronic non-communicable diseases associated with low-level inflammatory processes are a risk factor for infertility/subfertility in couples. Changes in the lifestyle and diet of infertile/subfertile couples can improve the inflammatory state and the level of oxidative stress of the couple, favoring fertility in couples in which at least one of the members has a chronic non-communicable disease associated with low-grade chronic inflammation [106,107,108,109,110].

### 3.2. Optimizing Couples Fertility—Precision Nutrition

Precision nutrition is based on a multidisciplinary approach, with the integration of different elements, some more traditional, such as eating habits and anthropometry, and others coming from new areas of knowledge, such as genetic variation, intestinal microbiota and predictive biochemistry. Nutritionists are increasingly aware of the relevance of personalized nutrition intervention in promoting couples’ health and fertility [111,112]. Precision nutrition uses information about individuals’ specific needs to predict their response to nutrition and lifestyle interventions with the goal of promoting health [113]. To assess an individual’s specific nutritional needs, it is necessary to carry out a genetic predisposition test for dietary and lifestyle factors as well as the subsequent identification of intermediate phenotypes for the highest genetic risk factors and the development of personalized dietary and lifestyle recommendations, typically using appropriate clinical nutrition software [114].

#### 3.2.1. Nutrigenetic Tests

Nutrigenetic tests are classified as predictive tests that analyze genetic variations resulting from the presence of SNPs in one or both alleles of genes with known function. These tests are based on the principle that individuals with risk genotypes are more susceptible to developing some chronic non-communicable diseases and are unable to meet their individual needs for nutrients and lifestyle parameters [115,116]. Nutrigenetic tests allow you to personalize dietary recommendations and assess predisposition to different individual nutritional needs. For example, some specific genotypes are, according to scientific evidence [117,118], of greater risk for a person’s sensitivity to lactose, caffeine or gluten. This type of test can also detect higher-risk genetic variations that predispose to changes in energy metabolism and nutrient concentrations, such as fatty acids and folates, as well as a greater risk of developing obesity or having greater specific needs for vitamins and minerals [116]. Therefore, nutrigenetic tests are a very useful tool for personalizing eating plans when analyzed in an integrated manner with other conventional parameters evaluated in precision nutrition consultations. This type of consultation has been increasingly sought after by couples with subfertility [13,14,119], as they are offered an individualized nutrition intervention and an integrative approach to the different parameters assessed in both members of the couple, which have an impact on their reproductive health [120,121,122].

#### 3.2.2. Predictive Biochemical Analyses

Biochemical analyses combined with nutrigenetic tests are essential for personalized nutrition interventions [123], as they allow the identification of intermediate phenotypes and facilitate personalized nutrition intervention. In this context, a variety of biomarkers present in biological samples can be included, which provide essential information about the metabolic state of an individual after their main genetic risks have been identified. For example, by knowing the plasma levels of homocysteine (an independent risk factor for cardiovascular disease) [124] and some B vitamins (B6, B9 and B12) after the identification of C677T polymorphism of methylenetetrahydrofolate reductase (MTHFR) in an individual without cardiovascular disease, it is possible to develop a targeted and preventive nutrition intervention by increasing the levels of B vitamins and decreasing the supply of methionine (homocysteine precursor).

#### 3.2.3. Gut Microbiota Tests

The microbiota has a profound impact on our health, influencing gene expression and affecting the composition of proteins (proteome), which regulate various metabolic functions [125]. Intestinal microbiota tests allow us to assess the presence of dysbiosis and identify the proportion of different bacterial strains that colonize the intestine. These types of tests have been used as one of the elements of precision nutrition with the aim of personalizing dietary recommendations, favoring the growth of beneficial bacteria and the consequent optimization of metabolic health [126]. The information obtained from intestinal microbiota tests makes it possible to direct nutrition intervention strategies using pre- and probiotics, with the aim of replenishing intestinal flora that promotes health and reduces the risk of developing chronic non-communicable diseases [127,128]. The presence of intestinal dysbiosis has been identified as a risk factor for infertility [129,130], especially in men. Optimizing intestinal health favors the fertility of couples, and the use of synbiotics/probiotics [131,132] is one of the strategies used within the scope of precision nutrition to modify the intestinal flora and end dysbiosis in one or both members of the couple.

#### 3.2.4. Use of Clinical Nutrition Software

Another essential component of precision nutrition is the use of clinical nutrition software that allows a complete analysis of the eating plans created as part of the individualized nutrition intervention [133]. With these programs, it is possible to assess whether the nutrition intervention and the respective individualized dietary plan meet the previously defined objectives based on the different elements that make up precision nutrition. This type of software allows the recording of genetic, biochemical, anthropometric and dietary data, among others, and with this information, a food plan based on the nutritional needs and assessed individual characteristics can be created [134]. After its elaboration, it is possible to check the adequacy of the plan for the previously defined energy needs as well as for macronutrients, their constituents and micronutrients. Usually, this type of software allows a more personalized intervention and monitorization through the installation of applications that define daily tasks and activities to comply with the eating plan and respective nutrition and lifestyle recommendations. The use of this type of software encourages adherence to the eating plan and facilitates more regular and closer monitoring of individuals seeking precision nutrition consultations [108,112].

### 3.3. Personalized Nutrition Interventions to Optimize Fertility

#### 3.3.1. Dietary Patterns

The effect of diet and lifestyle on couples’ fertility has been studied over the years [135,136]. Adopting different eating patterns influences the fertility of couples due to differences in nutrient intake that can have a positive or negative effect on female and male fertility [137,138,139,140,141]. Recent studies have highlighted the positive impact of adherence to Mediterranean-influenced dietary patterns on the reproductive health of couples [121,142,143]. This dietary pattern, rich in fiber, omega-3 fatty acids, vegetable proteins, vitamins and minerals, favors the consumption of vegetables, fruits, olive oil, complex carbohydrates, low-fat dairy products, poultry, fatty fish and a moderate intake of red wine. The Mediterranean diet promotes male fertility, in part due to its low levels of saturated and trans fats while providing optimal amounts of nutrients such as omega-3 fatty acids, antioxidants and vitamins. Adherence to this dietary pattern has been shown to be beneficial in improving motility and the number and quality of sperm [143,144,145]. The increase in sperm count has also been associated with a greater intake of vitamins, antioxidants and carotenoids [146,147]. Sperm motility and concentration seem to improve with the increasing sperm volume as the result of the consumption of fruits, cereals and vegetables [141]. In relation to female fertility, the pattern of the Mediterranean diet has been indicated as a factor that appears to reduce the risk of weight gain and insulin resistance, thus improving the chances of conception [110,148]. The Mediterranean diet has been recommended to couples in the preconception period during in vitro fertilization treatment [110,120,149,150]. Other dietary patterns have also been studied [137,151,152] by the scientific community, with evidence of the existence of adverse effects from standard Western food on couples’ fertility. This pattern is characterized by the consumption of foods high in refined carbohydrates, red meat and processed foods rich in saturated and trans fat. In this dietary pattern, the low consumption of fruits and fresh vegetables is also common. According to current knowledge [121], a diet rich in trans and saturated fatty acids as well as refined carbohydrates and added sugars can have harmful effects on female and male fertility. According to Grieger et al. (2018) [152], low fruit and high fast-food consumption in the preconception period is associated with a longer period until conception and an increased risk of infertility. The authors also found that small changes in food consumption can be beneficial for conception. In agreement, Silvestris et al. (2019) [15] concluded that dietary patterns with a high glycemic index, rich in animal protein, trans and saturated fatty acids may have a negative effect on fertility.

#### 3.3.2. Macronutrients

##### Carbohydrates

Glucose metabolism and insulin sensitivity are two factors that have a significant impact in male and female fertility [110,153,154]. Carbohydrate consumption with a high glycemic index favors insulin resistance, stress oxidative stress, the development of diabetes, dyslipidemia and metabolic changes that are closely linked to fertility [155,156,157]. Adopting a Western diet, rich in processed carbohydrates with a high glycemic index, has been associated with a decline in semen quality parameters [141]. In men, a diet with a high glycemic index can indirectly lead to fertility problems, inducing oxidative stress, a key factor in reducing sperm quality and increasing sperm risk of infertility. Hyperglycemia negatively affects the motility of spermatozoa and can result in hormonal and immunological disorders [138,158]. Alizadeh et al. (2017) [159] observed that the consumption of cookies and sugary drinks has a negative effect on the progressive motility of sperm, increasing the risk of azoospermia. The amount of carbohydrates ingested daily also seems to play a fundamental role in preventing infertility. In women, adopting a low-carbohydrate diet can significantly increase the likelihood of pregnancy and maintaining a regular menstrual cycle [120,160]. Additionally, low-calorie and low-glycemic diets have been associated with better reproductive outcomes, with a greater number of oocytes retrieved and live births in obese and infertile women undergoing in vitro fertilization, as well as higher rates of spontaneous pregnancy [160,161].

##### Protein

Adequate protein intake is another essential aspect of nutrition intervention to promote couples’ fertility [120]. Both the amount and type of protein in food can significantly influence fertility results in men and women [137]. A low-protein diet has been identified as a potential risk factor for male infertility, as it is responsible for a significant reduction in the weight of reproductive organs, such as testicles, the epididymis and seminal vesicles, along with a decrease in serum testosterone levels [141]. Additionally, it has been reported that the source and amount of protein consumed affects insulin sensitivity [123,162]. A higher protein intake can help achieve a balance between carbohydrate intake and insulin release and may be essential for the treatment of anovulatory infertility in women [163]. Animal protein consumption has been associated with an increased risk of infertility attributed to ovulation problems [15,120]. Inversely, the inclusion of vegetable protein in the diet has been shown to increase fertility in women over 32 [164]. These differences between protein types may be related to different effects on the secretion of insulin and type 1 insulin-like growth factor. Vegetable protein consumption appears to provoke a lower insulin response compared to animal protein intake. In a cohort study [164] carried out in women with no prior history of infertility, the risk of ovulatory infertility increased by 32% in women who increased their daily intake of protein of animal origin (e.g., red meat, chicken, turkey, processed meats and fish) while maintaining the caloric intake. The authors suggested that replacing natural sources of animal protein, especially chicken and red meat, with plant-based alternatives could potentially decrease the risk of infertility due to anovulation. The negative impact of meat consumption (especially processed meat) on fertility may be attributed to its high levels of saturated fats, trans fatty acids, preservatives and hormonal residues [120,165]. Current scientific evidence shows that trans fatty acids present in meat can affect sperm quality [141,166,167] and that the consumption of processed red meat is inversely related to the total count of motile sperm in the ejaculate [138]. Another aspect relevant to the differences found between the effect of animal and vegetable protein on the fertility of couples is the amino acid profile, as vegetable proteins contain a lower amount of sulfur amino acids (methionine and cysteine) compared to proteins of animal origin. These two amino acids and phenylalanine can influence sperm quality, decreasing their progressive motility in vitro [141]. Several studies [168,169,170] show that sperm parameters among vegan and non-vegan individuals are significantly different, with a beneficial effect for male fertility when diet is based on fruits and vegetables. Men on a vegan diet appear to have a significantly higher percentage of spermatozoa with high motility compared to those that consume products of animal origin [168]. Also, in an epidemiological case–control study [171] involving 30 men with reduced semen quality and 31 normozoospermia controls, it was observed that the dietary pattern between the two groups was different. Individuals in the control group consumed more vegetables and fruits, and those in the case group had a higher consumption of meat and yogurt [170]. Contrarily, Samimisedeh et al. (2023) [171] observed no association between the adherence to a vegetarian diet with semen quality and changes in sperm parameters. More studies are needed to confirm possible associations between a vegetarian diet and infertility and its determinants.

##### Fat

Lipids are macronutrients that play a crucial role in the fertility of men and women [121,172]. A diet with an adequate supply of acid fat is essential in preventing fertility problems. Both the excessive consumption and low intake of dietary fats promote the development of fertility problems, negatively affecting reproductive health [120]. Trans fatty acids (TFAs) are the ones that stand out the most due to their significant impact on female fertility [121,173], as they impair ovulation in women because of increased inflammatory markers and insulin resistance. By promoting the inflammatory process, TFAs increase the risk of DM2 and other metabolic disorders that can negatively affect fertility [174,175]. Hohos and Skaznik-Wikiel (2017) [176] showed that a high dietary fat intake was associated with changes in reproductive functions, influencing factors such as the duration of the menstrual cycle, reproductive hormone concentrations and embryo quality during medically assisted reproduction cycles. Beyond the fat profiles of food, what seems to be most impactful is the amount ingested. A study developed by Chavarro et al. (2007) [174], including 18,555 women who planned to become pregnant or who were already pregnant, showed that replacing 2% of energy from polyunsaturated or monounsaturated fatty acids with AGT doubled the risk of anovulatory infertility. Saturated fatty acids (SFAs) have also been identified as a factor promoting adverse effects on ovulation [173,177]. The effects of high SFA and trans intake on male fertility are also described in the scientific literature [141,167,178,179]. The presence of trans fatty acids in semen has been correlated with a decrease in its quality and a lower concentration of sperm in the ejaculate [138]. A high intake of SFA has been associated with reduced blood count and sperm concentration [180,181,182], as well as decreased sperm motility [181,183] and abnormal sperm morphology [184,185]. In addition to AGS and trans, the levels of plasma cholesterol also appear to be inversely associated with semen blood volume [186]. On the other hand, polyunsaturated fatty acids (PFAs) seem to bring reproductive benefits for both men and women [120]. It has been suggested that omega-3 fats are the most important component of sperm membranes due to their role in improving sperm motility, membrane fluidity and fertility potential [183,187]. According to the scientific literature, AGPs have a positive impact on sperm concentration, count and morphology [138], and the high consumption of omega-3 fatty acids appears to be associated with significantly better sperm [180]. In a meta-analysis with 16 clinical randomized trials carried out by Falsig et al. (2019) [188], it was possible to conclude that omega-3 supplementation is positively correlated with semen quality in men facing infertility problems. In another scientific work [189], which involved a double-blind randomized clinical trial in men undergoing fertility assessments, it was observed that individuals who received 1500 g/day of omega-3 fatty acid in the form of docosahexaenoic acid (DHA) for 10 weeks had better serum levels of DHA in seminal plasma and a reduction in the percentage of sperm with DNA damage. In women, the intake of omega-3 appears to have a protective effect on fertility. Thus, women of childbearing age with a high intake of omega-3 in their daily diet are more likely to become pregnant compared to those with omega-3 dietary intake values lower than those recommended by the European Food Safety Authority (EFSA) [190,191,192]. The results obtained by Moran et al. (2016) [191] allowed us to observe that the consumption of AGP was associated with pregnancy success in women with pre-obesity and obesity undergoing in vitro fertilization treatments. In general, and according to the scientific evidence described above, one could say that the intake of fats has a preponderant role in the design of diets, both in terms of quantity and in the type of fatty acids. SFA, trans and cholesterol appear to have a negative effect on male and female fertility [193]. In AGP, especially in omega-3 fatty acids, the effect appears to be protective when its consumption is daily and adequate [194,195].

#### 3.3.3. Micronutrients

##### Effect on Female Fertility

B Complex Vitamins

Vitamins and minerals have known effects on pregnancy and the preconception period. An inadequate intake of micronutrients is associated with female fertility problems [164,196]. Some vitamins such as B9, B12 and B6 have a known impact on women’s fertility [197]. The relationship of these three vitamins with fertility may be related to changes in homocysteine metabolism, an independent marker of cardiovascular disease [198,199]. The combination of high levels of homocysteine, together with insufficient levels of folic acid, represents a factor in the risk of recurrent miscarriage and has implications for women’s fertility [121]. Additionally, elevated homocysteine levels in ovarian follicle fluid can interrupt the interaction between the follicle and the sperm, leading to a reduction in the probability of fertilization [121]. Supplementation with vitamin B9 alone or especially combined with vitamin B12 in doses higher than those recommended for the prevention of birth defects can improve the chances of conception when taken before pregnancy [121]. In a cohort study carried out by Szymanski et al. (2003) [200] in women undergoing in vitro fertilization, it was observed that women who received folic acid supplementation before treatment had oocytes of better quality and a higher percentage of mature oocytes compared to those who did not take folic acid. According to scientific evidence, taking supplements of folic acid or multivitamins has been associated with greater embryo quality and a decreased risk of ovulatory infertility [164]. Folic acid supplementation before or during the early stages of pregnancy is associated with a decreased risk of miscarriage [63].

Vitamin D

Vitamin D plays a fundamental role in regulating female fertility [201,202,203,204]. Reproductive tissues have vitamin D receptors [13,14,121], and evidence suggests that vitamin D may have a positive effect on hormonal profiles and metabolic processes in women with polycystic ovarian syndrome and endometriosis [204,205]. In research carried out on women undergoing IVF treatment, those with adequate vitamin D levels had a pregnancy rate four times higher than that of women with insufficient concentrations of this vitamin [206]. Women with adequate levels of vitamin D had a higher rate of clinical pregnancy through in vitro fertilization (54.9%) than those that had inadequate levels (34.7%) [207]. In a case–control study in women [208], researchers compared baseline vitamin D levels in women who took between 12 and 14 months to conceive and women of the same age who conceived in less than 1 year, and no association was found between vitamin D levels and the time necessary to conceive. Gaskins and Chavarro (2018) [63] studied women who suffered 1–2 miscarriages (but without any history of infertility) without finding a significant relationship between basal serum levels or the deficiency of vitamin D and fecundability. In the same way, a meta-analysis developed by Amegah et al. (2017) [209] concluded that there was no significant association between insufficient vitamin D levels and the risk of spontaneous miscarriage. Thus, serum vitamin D levels appear to be a crucial factor for female fertility, especially in the presence of chronic inflammatory pathologies. However, more studies are needed to increase the consistency in the causal relationship between serum vitamin D levels and female fertility.

Minerals

There are some minerals to consider in relation to female fertility, one being iodine, as it affects the function of the thyroid gland, which is crucial for fertility [210]. In a study involving 501 women with moderate to severe iodine deficiency, there was a delay in pregnancy, with a lower probability (−46%) of conception per cycle compared to women without iodine deficiency [121]. Insufficient levels of zinc have also been linked to reduced pregnancy rates and cycles, as well as abnormal menstrual periods [14]. Low zinc and selenium serum levels were correlated with a one-month delay in the time needed to get pregnant, and low levels of selenium and copper have been associated with an increased risk of infertility [152]. Selenium plays a fundamental role in the body as an antioxidant, participating in a reduction in the level of oxidative stress associated with the inflammatory process and potentially affecting the growth and maturation of oocytes [121]. An adequate intake of selenium in healthy women and supplementation in pregnant women can potentially improve fertility [211,212].

Antioxidants

Oxidative stress has a relevant role in female reproduction [213], more specifically in ovulation, endometrial decidualization, menstruation, oocyte fertilization and the development and implantation of an embryo in the uterus [214]. A high level of oxidative stress promotes the appearance of disorders in reproduction in women and consequently triggers gynecological diseases that can end in infertility [215]. The action of antioxidants that block oxidizing species is crucial for maintaining an adequate level of oxidative stress, which is essential in the metabolism of oocytes, the maturation of the endometrium by the activation of antioxidant signaling pathways and the hormonal regulation of vascular action [216]. Their function as the cofactors of key enzymes in differentiation and cell development or of antioxidant enzymes is another way in which antioxidants work in the body. An inadequate intake of antioxidants can lead to a deficit of these compounds in the body and promote an environment that is not conducive to conception. Daily supplementation with some antioxidants can improve female fertility and improve the results of assisted reproduction techniques [214]. Some of the most powerful antioxidants include vitamin C, vitamin E and vitamin A [121]. Taking vitamin E supplements appears to be associated with a shorter time to conception among women over 30 years old, and women under 35 years old who consumed beta-carotene and vitamin C had a shorter time to conception [217]. The regular intake of ascorbic acid during pregnancy can promote hormone production in the human placenta, which plays a crucial role in supporting the gestation process [218,219]. Scientific evidence on the benefits of the regular and adequate consumption of antioxidants with the aim of improving reproductive outcomes is still insufficient and inconclusive, but studies that may establish a more consistent relationship between some antioxidants (e.g., vitamins C, E, A, carotenoids) and female fertility are being developed.

##### Effect on Male Fertility

B Complex Vitamins

In men, low plasma levels of vitamin B12 and B9 have been associated with infertility [220]. Serum vitamin B12 levels have been correlated with sperm concentration, motility, morphology and DNA damage [221]. Low serum concentrations of vitamin B12 also appear to be related to a greater risk of testosterone deficiency and impaired androgenic profiles, which affect spermatogenesis and, consequently, fertility [222]. Although some studies have not demonstrated any relationship between folate intake and semen quality, others suggest a positive effect of folate intake on male reproductive health [223]. The negative effects that low B9 levels have on sperm DNA, oxidative stress and apoptosis favor lower sperm counts [57]. Several clinical trials have shown that B9 supplements favor an increase in sperm concentration and motility [224,225,226,227]. Additionally, folate supplementation has been shown to have positive effects on male fertility, such as a reduction in sperm lesions, an increase in sperm density, normal sperm morphology, improvements in overall semen quality and a decrease in infertility rates [57].

Vitamin D

Vitamin D may also have an impact on fertility in men, possibly increasing sperm motility, viability and fertilization capacity [57,228]. Additionally, calcium plays a role in regulating sperm motility and has an impact on processes such as hyperactivation, sperm capacitation and acrosome reaction, which facilitates the penetration of sperm into the oocyte [142,229]. Furthermore, it can also have a positive effect on the morphology, motility, maturation and general function of sperm [57].

Minerals

Regarding minerals, the effect of zinc and selenium on male fertility is described by some authors [230,231]. Having sufficient levels of zinc in plasma provides protective benefits due to its antioxidant properties [138]. Maintaining adequate levels of zinc in semen is vital for several aspects of reproductive health, such as steroidogenesis, sperm production, testicle development and the preservation of sperm’s normal function, morphology and cell count [231]. Optimal levels of zinc in semen are associated with high concentrations of sperm in the ejaculate as well as greater sperm viability and motility and an increase in antioxidant activity [120]. Another element to consider is selenium, as it has a protective effect against oxidative stress in sperm DNA and improves sperm viability and motility [120]. Türk et al. (2014) [229] showed in their scientific work that selenium concentrations in the semen of infertile men were significantly lower than those found in the semen of healthy individuals. However, selenium insufficiency or excess can contribute to fertility problems and the parameters of abnormal semen [138].

Antioxidants

Vitamins E and C have powerful antioxidant properties and play an essential role in preventing infertility [232]. An increase in the dietary intake of vitamin C has been associated with decreased sperm DNA damage [57]. On the other hand, lower levels of seminal vitamin C are linked to greater sperm DNA fragmentation in infertile men [57]. Supplementation with vitamin E decreases lipid peroxidation levels in sperm and increases their motility [57,233]. Another important compound in terms of antioxidant capacity and the effect it can have on male fertility is lycopene, a carotenoid found in red fruit and vegetable pigment. This carotenoid has a crucial role in decreased lipid peroxidation and DNA damage as well as improved immune function, sperm count and sperm survival rates [234]. Greater lycopene consumption is positively associated with the normal morphology of sperm [138]. A diet rich in antioxidants can also greatly improve fertility, with positive effects on semen parameters and an increase in the rate of fertility of 4 to 18 times [120,159]. In men with compromised fertility, lower concentrations of antioxidants have been observed when compared to fertile men [120]. According to the study by Martin-Hidalgo et al. (2019) [235], oral supplementation with antioxidants can improve the parameters of semen quality and is associated with a decrease in DNA damage. The results of this scientific work allowed us to conclude that just three months of supplementation with vitamins E and C, carnitine and ubiquinol can lead to improvements in sperm density and motility [235] and a reduction in the percentage of abnormal sperm [138]. Thus, the daily consumption of fruits and vegetables, foods rich in antioxidants, is associated with a higher percentage of motile sperm in fertile and infertile men and improving semen quality and fertility [120].

## 4. Conclusions

The relationship between nutrition and reproductive health has expanded significantly over the last decade, highlighting both the harmful effects of inadequate nutrition and the benefits of a balanced diet on fertility results. Conditions like obesity, type 2 diabetes mellitus, metabolic syndrome, malnutrition and essential nutrients can compromise reproductive health and reduce the chances of conception. Adherence to a healthy diet that favors foods rich in macro- and micronutrients with beneficial effects on male and female fertility should be prioritized as well as the optimization of the nutritional status of both members of the couple. The multidisciplinary approach to precision nutrition, with the use and integration of different tools (e.g., nutrigenetics, biochemical predictive tests, intestinal microbiota tests and the use of clinical nutrition software) will certainly be of great use to couples with subfertility, as it improves reproductive health and conception hypotheses.

## Figures and Tables

**Figure 1 nutrients-17-00103-f001:**
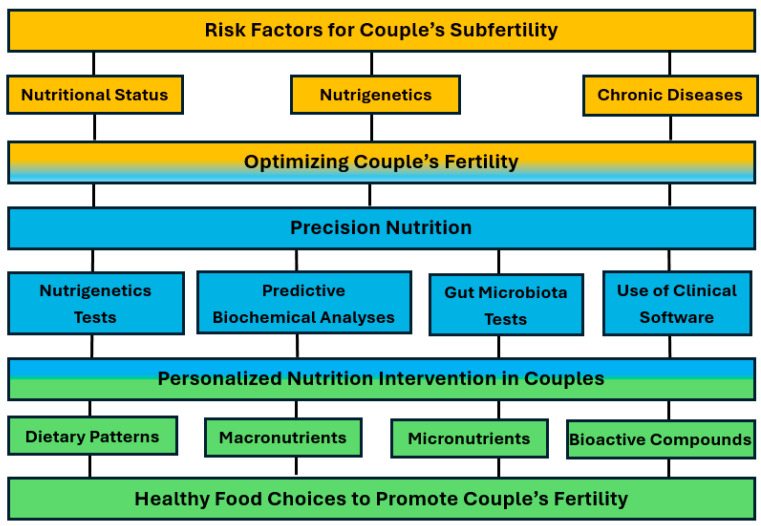
Risk factors and personalized nutrition intervention in couples with subfertility.

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
