# Peer review of "The Contribution of Precision Nutrition Intervention in Subfertile Couples"

_nutrients, 2024, doi:10.3390/nu17010103_

Round 1
Reviewer 1 Report
Comments and Suggestions for Authors
Interesting review about a currently highly relevant topic.
Major comments:
- Do the authors feel that the key words are sufficient? By using the terms “couple’s infertility” / “couple’s subfertility” one might have missed several relevant studies, since many studies focus on male/female in-/subfertility only.
- Methodology: It should be defined in detail whether the search terms were used for the whole review or only for the Section on “Precision Nutrition” and “Personalized Nutritional Interventions” etc.
- Regarding the discussion about obesity etc., there is a nice review about the influence of weight reduction on fertility outcomes: Hoek A, Wang Z, van Oers AM, Groen H, Cantineau AEP. Effects of preconception weight loss after lifestyle intervention on fertility outcomes and pregnancy complications. Fertil Steril. 2022 Sep;118(3):456-462.
- The authors could mention in the section about PCOS that this paragraph provides only a short overview. I suggest to add a few references where the reader can find more information.
- Vitamin D: several relevant references are not included, for example: Arab A, Hadi A, Moosavian SP, Askari G, Nasirian M. The association between serum vitamin D, fertility and semen quality: A systematic review and meta-analysis. Int J Surg. 2019 Nov;71:101-109. Thus, I would like to challenge the key words.
- Antioxidants: Same comment as above, I miss the following reference (for example): Showell MG, Mackenzie-Proctor R, Jordan V, Hart RJ. Antioxidants for female subfertility. Cochrane Database Syst Rev. 2020 Aug 27;8(8):CD007807. doi: 10.1002/14651858.CD007807.pub4.
- Conclusion: this section should focus a little more on the precision nutrition. I consider this the main focus of the present review rather than general statements about a (healthy) life style and fertility. Moreover, differences between men and women might be of interest.
Author Response
Answer to Reviewers
Revision 1
Dear Reviewer of Nutrients
The authors are grateful for the review and respond to the comments of the Reviewers.
Major comments:
- “Do the authors feel that the key words are sufficient? By using the term “couple’s infertility” / “couple’s subfertility” one might have missed several relevant studies, since many studies focus on male/female in-/subfertility only.”
Author’s response: The authors used the individual and combination of the following keywords: “genetic variation,” “nutrigenetics,” “precision nutrition,” "couple's subfertility," "couple's infertility." The use of these methodology resulted in several articles that focused only on male or female in/subfertility. As can be seen throughout the article, these studies were included in the results, which are presented with differentiation in relation to the main issues of subfertility for men and women.
- “Methodology: It should be defined in detail whether the search terms were used for the whole review or only for the Section on “Precision Nutrition” and “Personalized Nutritional Interventions” etc.“
Author’s response: Five initial keywords described in the previous point were used to create the methodology. Some allowed obtaining results associated with the description of risk factors for infertility/subfertility and others were used to write the results of point 2 “Optimizing Couples fertility - Precision Nutrition” and point 3 “Personalized nutritional intervention to optimize fertility”. The bibliographic references of the articles obtained as result of the initial research were also read and analyzed. Of these references, those considered relevant were also used in this narrative review of the scientific literature.
- “Regarding the discussion about obesity etc., there is a nice review about the influence of weight reduction on fertility outcomes: Hoek A, Wang Z, van Oers AM, Groen H, Cantineau AEP. Effects of preconception weight loss after lifestyle intervention on fertility outcomes and pregnancy complications. Fertil Steril. 2022 Sep;118(3):456-462.”
Author’s response: The authors would like to thank the reviewer for his suggestion. These are very interesting and recent bibliographical references, however the topic of the influence of nutritional status was addressed separately between men and women with the inclusion of around 40 references from various original articles and some narrative and systematic reviews with a temporal range.
- “The authors could mention in the section about PCOS that this paragraph provides only a short overview. I suggest adding a few references where the reader can find more information.”
Author’s response: The authors would like to thank the reviewer for his suggestion, however the topic PCOS was prepared with 12 references from various original articles and some narrative and systematic reviews with a temporal range. This article aims to highlight the importance that the presence of various chronic diseases can have on the subfertility of couples without delving too deeply into each of these pathologies.
- “Vitamin D: several relevant references are not included, for example: Arab A, Hadi A, Moosavian SP, Askari G, Nasirian M. The association between serum vitamin D, fertility and semen quality: A systematic review and meta-analysis. Int J Surg. 2019 Nov;71:101-109. Thus, I would like to challenge the key words.”
Author’s response: The authors would like to thank the reviewer for his suggestion. The article was added the references.
- “Antioxidants: Same comment as above, I miss the following reference (for example): Showell MG, Mackenzie-Proctor R, Jordan V, Hart RJ. Antioxidants for female subfertility. Cochrane Database Syst Rev. 2020 Aug 27;8(8):CD007807. doi: 10.1002/14651858.CD007807.pub4.”
Author’s response: The authors would like to thank the reviewer for his suggestion. The article was added the references.
- “Conclusion: this section should focus a little more on the precision nutrition. I consider this the main focus of the present review rather than general statements about a (healthy) life style and fertility. Moreover, differences between men and women might be of interest.”
Author’s response: The authors consider that the conclusion should be simple and comprehensive. The conclusion drawn up and described below has a first sentence that highlights the importance of the topic and the contribution that nutrition can have in improving a couple's fertility. It then cites the importance of assessing the presence of some risk factors as causes for infertility or subfertility and highlights the relevance of adhering to a healthy diet and adequate weight management to optimize couples' fertility. It ends by highlighting precision nutrition as an integrative approach that can be of great use in improving the chances of conception in subfertile couples with no known etiology.
“The relationship between nutrition and reproductive health has expanded significantly over the last decade, highlighting both the harmful effects of inadequate nutrition and the benefits of a balanced diet on fertility results. Conditions like obesity, type 2 diabetes mellitus, metabolic syndrome, malnutrition and essential nutrients can compromise reproductive health and reduce the chances of conception. Adherence to a healthy diet that favors foods rich in macro and micronutrients with beneficial effects on male and fe-male fertility should be privileged, as well as the optimization of the nutritional status of both members of the couple. The multidisciplinary approach to precision nutrition, with the use and integration of different tools results (e.g. nutrigenetics, biochemical predictive tests, intestinal microbiota tests and the use of clinical nutrition software), will certainly be of great use to couples with subfertility, as it improves reproductive health and conception hypotheses.”

Reviewer 2 Report
Comments and Suggestions for Authors
The manuscript titled “The Contribution of Precision Nutrition Intervention in Subfertile Couples” was done by Jéssica Monteiro et al., and there are some concerns about this review.
Major concerns
1. The title mainly focus on the subfertility, but in the manuscript some summarize focus on the infertility. In fact, subfertility and infertility are two conception.
2. Although more reference was cited to summarize the nutrition intervention in sub- or infertility couples, but more about the metabolism to sub- or infertility.
3. In part of Obesity, overweight and underweight in women, the obesity is not always related with the nutrient, but might correlation with the metabolism. In addition, the obesity is not only related with the ovulation, and obesity might affect growth and development of ovarian follicle.
4. The review is disorganisation, and it should be concise and rearrange, such as showed with the figures and tables.
Minor revision
1. In line 109, “who associated a gain of one unit in BMI above 29 kg/m2”, 2 revised into superscript. The same problem in line 138, 160.
2. Line 213, “Ramirez Torres et al. (2000] [74]”, please to revise the bracket.
3. In line 502-503, delete retardant space.
Author Response
Answer to Reviewers
Revision 1
Dear Reviewer of Nutrients
The authors are grateful for the review and respond to the comments of the Reviewers.
Reviewer 2
Major concerns
- The title mainly focus on subfertility, but in the manuscript some summarize focus on the infertility. In fact, subfertility and infertility are two conceptions.
Author’s response: It is a very pertinent observation, and the authors thank the reviewer.
The rationale for choosing the title is as follows: The relationship between infertility and nutrition has been studied for decades and in the past, it was thought that everything was infertility, as there was no differentiation between the terms subfertility and infertility.
According to the current definition of the American Society for Reproductive, the title highlights the role that nutrition can play in subfertility but not in all types of infertility. Couples are considered subfertile after no known etiology is identified in either partner, suggestive of impaired reproductive capacity. Therefore, the authors consider that the title of the article is the one that best suits the review presented, although it is necessary to mention the term infertility because it is the way it is described in bibliographic references.
(https://www.asrm.org/practice-guidance/practice-committee-documents/denitions-of-infertility/)
- “Although more reference was cited to summarize the nutrition intervention in sub- or infertility couples, but more about the metabolism to sub- or infertility.”
Author’s response: The authors cited more than 240 bibliographic references that included original articles, narrative and systematic reviews and meta-analyses of scientific literature. During the preparation of this review, careful standards were maintained to check the quality and diversity of the journals to be included. One hundred and three references were included in the topic “Personalized nutritional intervention to optimize fertility”, a similar value to the topic “Risk factors for couple subfertility” in which around 100 bibliographic references were used.
“In part of Obesity, overweight and underweight in women, the obesity is not always related with the nutrient but might correlation with the metabolism. In addition, obesity is not only related with the ovulation, and obesity might affect growth and development of ovarian follicle.”
Author’s response: Obesity may only be correlated with metabolism, however this review article only addresses nutrient-related obesity in women, since if there are causes for metabolic obesity in women, these must be evaluated and identified as possible causes for infertility. feminine. In this article, the focus is the importance of maintaining an adequate nutritional status (underweight or obesity) to enhance the couple's fertility.
- “The review is disorganisation, and it should be concise and rearrange, such as showed with the figures and tables.”
Author’s response
The review article is organized into:
- Risk factors for couple subfertility
- Optimizing Couples fertility - Precision Nutrition
- Personalized nutritional intervention to optimize fertility
A small adjustment was made in relation to point 2. to match the figure.
Minor revision
- “In line 109, “who associated a gain of one unit in BMI above 29 kg/m2”, 2 revised into superscript. The same problem in line 138, 160.”
Author’s response: Revision was made.
Line 109: “The researchers Van Der Steeg et al. (2007) [20] that evaluated whether obesity affected the chance of a spontaneous pregnancy, found that the probability of a spontaneous pregnancy linearly declined with a BMI over 29 kg/m2)”.
Line 138: “Chavarro et al. (2010) [41] showed that men with a BMI ≥ 35 kg/m2 showed a decrease in ejaculate volume and a lower total sperm count”.
Line 160: “Previously, Jensen et al. (2004) [51] carried out a study on 1558 young Danish men observed that participants with a BMI <20 kg/m2 had a significant reduction in sperm concentration and total sperm count”
- “Line 213, “Ramirez Torres et al. (2000] [74]”, please to revise the bracket.”
Author’s response: Revision was made.
- “In line 502-503, delete retardant space.”
Author’s response: Revision was made.

Round 2
Reviewer 1 Report
Comments and Suggestions for Authors
Well revised.
Author Response
The authors would like to thank the review carried out, which contributed to improving the quality of the scientific work.
Reviewer 2 Report
Comments and Suggestions for Authors
The manuscript titled " The Contribution of Precision Nutrition Intervention in Subfertile Couples" was done by Jéssica Monteiro et al.
The author have been all my concerns.
Author Response

(The authors gave the same response as above.)
